# Structure–Property Relationships of Pure Cellulose and GO/CEL Membranes Regenerated from Ionic Liquid Solutions

**DOI:** 10.3390/polym11071178

**Published:** 2019-07-12

**Authors:** Czesław Ślusarczyk, Beata Fryczkowska

**Affiliations:** Institute of Textile Engineering and Polymer Materials, University of Bielsko-Biala, Willowa 2, 43-309 Bielsko-Biala, Poland

**Keywords:** cellulose, graphene oxide, ionic liquid, membrane, transport properties, heavy metals, porous structure, SAXS, WAXS

## Abstract

Two types of cellulose membranes were produced by a classical wet phase inversion method from a solution of the polymer in 1-ethyl-3-methylimidazolium acetate (EMIMAc) by coagulation in water and selected primary alcohols. The first type were membranes made from pure cellulose (CEL). The second type were membranes obtained by adding nanosized graphene oxide (GO) to the cellulose solution. The process of precipitation and selection of the coagulant affected the structure of the membranes, which in turn affected their usability and applicability. The results of the presented studies show that the physicochemical properties of the coagulant used (e.g., molecular mass and dipole moment) play important roles in this process. It was found that both the content and dimensions of the pores depended on the molecular mass of the coagulant used. It was also found that the dipole moment of coagulant molecules had a large influence on the volume content of the pores (e.g., the 1-octanol (Oc) membrane had a dipole moment of 1.71 D; *Φ* = 1.82%). We investigated the effect of the type of coagulant on the porous structure of CEL membranes and how this affected the transport properties of the membranes (e.g., for the distilled water (W) membrane, *J_v_* = 5.24 ± 0.39 L/m^2^ h; for the Oc membrane, *J_v_* = 92.19 ± 1.51 L/m^2^ h). The paper presents the results of adding GO nanoparticles in terms of the structure, morphology, and transport properties of GO/CEL membranes (e.g., for composite membrane F (containing 20% GO), *J*_v_ = 40.20 ± 2.33 L/m^2^ h). In particular, it describes their extremely high ability to remove heavy metal ions.

## 1. Introduction

Cellulose is one of the oldest natural polymers. It is renewable, biodegradable, and can be derivatized to yield various useful products, such as paper, films, fibers, membranes, hydrogels, aerosols, microspheres, and beads [1,2]. Hence, cellulose is already the basis of a large industry, and although it cannot compete with petrochemistry in terms of costs, it seems very likely that it will be the main chemical resource of the future, as it will still be available when other substances become increasingly scarce due to exhausted reserves or environmental difficulties.

Cellulose is a linear polymer linked with stable glycosidic bonds accompanied by intra- and intermolecular hydrogen bonds. The presence of these bonds as well as the presence of crystalline areas in cellulose makes this polymer insoluble in water and in most organic solvents [3]. Many systems capable of dissolving the polymer have been developed, including NaOH/CS_2_, dimethyl sulfoxide and tetrabutylammonium fluoride mixture (DMSO/TBAF), LiCl/dimethylacetamide (DMAc), and N_2_O_4_/*N*,*N*-dimethylformamide (DMF) [4,5], whose disadvantages are volatility, toxicity- or flammability.-. Another system used to dissolve cellulose is a mixture of *N*-methylmorpholine-*N*-oxide and water (NMMO/H_2_O), which—unlike the previously mentioned systems—is environmentally friendly and cost-effective [6].

Recent cellulose research has paid a great deal of attention to ionic liquids, often referred to as “green” solvents. Due to their biodegradability [7] and low toxicity [8], they are considered to be environmentally friendly solvents, and may eventually replace traditional systems that are capable of dissolving cellulose [9,10]. Ionic liquids are a type of organic salt with a liquid phase below boiling temperature. Their cationic component determines their chemical stability, while their anionic part plays an important role in chemical reactions [11,12,13]. The use of ionic liquids for making cellulose solutions renders several advantages such as a lower dissolving temperature, lower or even zero toxicity, better thermal stability, and easy solvent recovery [14,15].

Swatloski et al. [16] were the first to describe the dissolution of cellulose in 1-butyl-3-methyl imidazolium chloride, and they found that the chemical structure of the imidazole cation in the ionic liquid has a strong influence on the viscosity of the obtained polymer solutions. Kosan et al. [11] investigated the properties of various ionic liquids, but the most interesting turned out to be 1-butyl-3-methyl imidazolium acetate, which enabled the preparation of a 13.5% cellulose solution. This liquid has even more advantages, such as a low melting point, high boiling point, and good thermal stability [17]. In their studies on the dissolution of cellulose in ionic liquids, Ding et al. used a 5% solution of microcrystalline cellulose which they dissolved in 1-ethyl-3-methylimidazolium acetate (EMIMAc). Mahadeva et al. [18] used a 2% solution of cellulose for membrane molding, and coagulation was carried out in water, methanol, a mixture of methanol with water (30:70), and isopropyl alcohol. Sun et al. [19] reported that the 5% solution of cellulose in EMIMAc was coagulated in the acetone–water mixture (1:1 *v/v*). Hermanutz et al. [15] used 10%, 12%, and 16% cellulose solutions in EMIMAc to produce cellulose fibers by coagulation in water. Livazovic et al. [20] described the preparation of cellulose solutions at concentrations of 2.0%, 5.0%, and 10% cellulose in 1-ethyl-3-methylimidazolium acetate by heating at 80 °C for 24 h. The solutions obtained were molded with a casting knife with an adjustable thickness fixed at 0.15 mm, and the resulting films were coagulated in distilled water.

Cellulose dissolved in ionic liquid can easily be precipitated using polar solvents such as water, acetone, ethanol, dichloromethane, acetonitrile, or by using a mixture of these solvents [14,21], to obtain useful products. The process of precipitation and the selection of the coagulant affects the coagulated polymer structure.

Membranes are among the products made from regenerated cellulose which deserve special attention. These materials are widely used in a variety of membrane separation techniques, such as the dialysis, ultrafiltration-, and purification of mixtures, and are used in sewage treatment plants in the process of water treatment, in the chemical and food industries, as well as in environmental protection, biochemistry, and medicine [22,23]. According to literature reports, the factors affecting the properties of membranes include the type of coagulating bath, its concentration, and the time of film formation, among others [22].

However, most often the membranes described by the literature are composites. The authors describe the process of obtaining a cellulose/iron oxide bionanocomposite by the mixing blends technique [24,25]. Rac-Rumijowska et al. described a method for the preparation of cellulose-based fibrous composites with the addition of silver nanoparticles [26]. Yue’s group constructed anti-bacterial composites based on paper pulp with the addition of silver ions [27]. Other researchers made composite membranes based on cellulose with the addition of chitosan and carbon nanotubes [28]. Graphene oxide (GO) has also been used for the preparation of composites with cellulose [29,30,31]. Yadav et al. used cellulose derivatives [32,33] to obtain GO-containing composites.

This paper presents a study on the production of cellulose membranes by the classical wet phase inversion method. EMIMAc was selected as the solvent, since—according to Sun et al. [19] and also confirmed by studies of Gupta et al. [34]—this ionic liquid breaks the hydrogen bonds in cellulose most easily. The cellulose dissolved in ionic liquids can be coagulated in polar solvents [35]. In our experiment, water and primary alcohols differing in polarity such as methanol, ethanol, 1-propanol, 1-butanol, 1-hexanol, and 1-octanol were used for the coagulation of cellulose membranes. In this paper, we describe the study of two types of membranes. The first type were membranes made from pure cellulose (CEL). The second type were membranes obtained by adding nanosized graphene oxide (GO) to the cellulose solution. A method of combining the cellulose solution with the dispersion of GO additive was developed to obtain a homogeneous solution from which the composite membranes (GO/CEL) were then formed using the phase inversion method by coagulation in distilled water.

The process of precipitation and selection of the coagulant affects the structure of the membranes, which in turn affects their usability and applicability. The effect of the type of coagulant on the porous structures of CEL membranes was investigated, in addition to how this affects the transport properties of the membranes. For GO/CEL membranes, this paper presents the results of investigations of the effect of adding GO nanoparticles on the structure, morphology, and transport properties of these membranes. In particular, it describes their extremely high ability to remove heavy metal ions.

## 2. Materials and Methods

### 2.1. Materials

Cellulose (long fibers), ionic liquid (EMIMAc), and graphite powder (<20 μm) were purchased from Sigma-Aldrich (Poznań, Poland). NaNO_3_, 98% H_2_SO_4_, KMnO_4_, 30% H_2_O_2_, *N,N*-dimethylformamide (DMF), Co(NO_3_)_2_, Ni(NO_3_)_2_, Pb(NO_3_)_2_, and ZnCl_2_ were purchased from Avantor Performance Materials Poland S.A. (Gliwice, Poland). All chemicals were used without further purification.

### 2.2. Preparation of the Cellulose Solution

Initially, a 5% solution of cellulose in the ionic liquid (EMIMAc) was prepared. For this purpose, adequate amounts of cellulose and ionic liquid were combined to obtain solutions with concentrations of 2.5%, 5.0%, 7.5%, and 10% by weight. The solutions were then mixed thoroughly and heated in a laboratory microwave oven at three intervals of 5 s, keeping the temperature of the mixture below approximately 40 °C. Previously, it was observed that very accurate mixing at intervals between heating prevents the local overheating of cellulose and its degradation [36]. The obtained cellulose solutions were allowed to de-aerate for 24 h. The viscosity of the prepared cellulose solutions was measured at room temperature by means of a Myr V2-L rotary viscosity gauge, equipped with an L3 spindle and temperature sensor (Especialidades Medicas Myr, SL, Tarragona, Spain) (Figure 1).

Looking at Figure 1, we can see that the viscosities of the obtained cellulose solutions in EMIMAc increased drastically with their concentration. Above 10% wt. concentration, the cellulose solutions formed a gel that prevented further dissolution of cellulose. As a result, 5% cellulose solution was selected for further investigation. A solution of this concentration was also used by Ding et al. for their research [37].

### 2.3. GO Synthesis and Preparation of GO/DMF Dispersion

GO synthesis was carried out in our laboratory using the modified Hummers’ method [38], as described in our earlier work [39]. Wet GO was dried in a laboratory drier at 60 °C to obtain a brown precipitate which was then dispersed in DMF in an ultrasonic bath to obtain a 5.2% GO/DMF dispersion (Figure 2).

### 2.4. Pure Cellulose Membrane and GO/CEL Membrane Formation

Cellulose membranes were prepared using the phase inversion method. For this purpose, a 5% cellulose solution in EMIMAc was poured onto a level, clean glass plate. Then, a polymeric film was formed using a casting knife with an adjustable thickness fixed at 0.2 mm. After that, the plate and the cellulose solution distributed over it were transferred to a coagulant bath at room temperature and kept in it until the membrane peeled off the glass. The following solvents were used in coagulation solutions: distilled water (W), methanol (Me), ethanol (Et), 1-propanol (Pr), 1-butanol (Bu), 1-pentanol (Pe), 1-hexanol (He), and 1-octanol (Oc). In the presence of water, cellulose membranes coagulated rapidly (10–30 min). Then the membranes were immersed in alcohols containing one to four carbon atoms in their molecules. The coagulation of cellulose solutions in higher alcohols was very slow, and it took 1 day (1-pentanol), 4 days (1-hexanol), and 5 days (1-octanol) for the membranes to separate from the glass plate, indicating the finalization of the process. After the completion of this process, the membranes - Pe, He, and Oc were repeatedly rinsed, alternately with distilled water and methanol, until the characteristic odor of the alcohol disappeared. Then, they were air-dried at room temperature. In our laboratory, a thin layer of polyester fabric followed by layers of tissue paper were used to separate the cellulose membranes. This prevented the cellulose from sticking to the filter paper and facilitated drying. Finally, the dried membranes were loaded onto a glass plate to prevent membrane ripple (Figure 3).

In order to prepare solutions for forming the GO/CEL composite membranes, adequate amounts of cellulose and ionic liquid were first weighed (Table 1), and cellulose solutions were prepared as described above. Appropriate amounts of 5.2% GO/DMF solution (Table 1) were then added to the cellulose solutions, thoroughly mixed, and sonicated for 15 min. The obtained GO/CEL solutions were allowed to de-aerate for 24 h. GO/CEL composite membranes were prepared using the phase inversion method, as was done for the membranes made from pure cellulose.

### 2.5. Measurements of Water Flux

The transport properties of the formed membranes were tested using a Millipore Amicon 8400 ultrafiltration (UF) cell with a 350 mL capacity and a 7.6 cm membrane diameter that was equipped with an equalizing tank with an 800 mL capacity (Merck Millipore, Burlington, MA, USA). First, dry membranes were immersed in distilled water for 1 h. Then, they were treated with room-temperature distilled water) for an additional 2 h under a pressure of 0.2 MPa to improve the membrane stability. UF tests were performed at operational pressures of 0.1, 0.15, or 0.2 MPa. Permeate flux (*J_v_*) was calculated using the following Equation (1) [40]:(1)$$J_{v}=\frac{Q}{A\;t}$$
where *J_v_* is the water flux (L/m^2^ h), *Q* is the permeate volume (L), *A* is the effective membrane area (m^2^), and *t* is the permeation time (h).

### 2.6. Measurements of Rejection

Standard solutions of Pb(NO_3_)_2_ with a concentration of 60 mg/L and NaCl, Co(NO_3_)_2_, Ni(NO_3_)_2_, and ZnCl_2_ with concentrations of 6 mg/L were prepared to study the separation properties of the produced GO/CEL membranes. In addition, a mixture of synthetic wastewater containing 6 mg/L Co(NO_3_)_2_, Ni(NO_3_)_2_, ZnCl_2_, and 60 mg/L Pb(NO_3_)_2_ was prepared.

Then, 200 mL of successive standard solutions were added to the UF cell with the test membrane and the stirrer (stirring of the feed solution avoids fouling). The permeation process was carried out at a working pressure of 0.2 MPa, and 20 mL doses of permeate were tapped, simultaneously measuring the time taken for the permeate discharge to from the test tank. The permeate flux (*J_v_*) was calculated using Equation (1), assuming that *Q* was the permeate volume (specific test solution).

The ion concentrations of the subsequent metals were determined using atomic absorption spectrometry (AAnalist 100 AAS, Perkin-Elmer International Rotkreutz Branch P.O., Rotkreutz, Switzerland), and the rejection coefficient (*R*) was calculated using Equation (2) [40]:(2)$$R=\left(1-\frac{C_{p}}{C_{f}}\right)100\%,$$
where *R* is the rejection performance of the membrane (%), and *C_p_* and *C_f_* are the concentrations of metal ions in the permeate and feed solution (mg/L), respectively.

### 2.7. Analytical Methods

Small-angle X-ray scattering (SAXS) measurements were carried out with the compact Kratky camera equipped with the SWAXS optical system of HECUS-MBRAUN (Graz, Austria). The Cu target X-ray tube operating at U = 30 kV and I = 10 mA was used as a radiation source (λ = 0.154 nm). The SAXS data were collected as a function of the scattering vector, q = (4π/λ) sin θ, where 2θ is the scattering angle. The moving slit method [41] was applied for determination of the transmittance factor of the sample. The sample holder background was subtracted from the SAXS curves, and the next curves were corrected, taking both the sample thickness and transmittance into consideration. The data were converted to absolute intensities with a calibrated Lupolen^®^ (polyethylene) standard [42].

Wide-angle X-ray scattering (WAXS) investigations were carried out with a URD-65 Seifert (Germany) diffractometer. Cu Kα radiation was used at 40 kV and 30 mA. Monochromatization of the beam was obtained by means of a nickel filter and a graphite crystal monochromator placed in the diffracted beam path. A scintillation counter was used as a detector. Investigations were performed in the angle range of 3° to 40° in steps of 0.1°.

Observations of the – morfology of membranes surface and cross-section were carried out using a JSM-5500 LV JEOL (Tokyo, Japan) scanning electron microscope (SEM). All samples were coated with a layer of gold in a JEOL JFC 1200 vacuum coater at 3 × 10^−5^ Tr.

## 3. Results and Discussion

### 3.1. Pure Cellulose (CEL) Membranes

#### 3.1.1. Porous Structure of CEL Membranes

The porous structure of membranes is one of their most important features, as it determines their usability and applications. This structure is mainly formed at the coagulation stage; therefore, the type of coagulant used has an influence on it [18,19]. In this paper, an analysis of the porous structure of the CEL membrane was carried out by the SAXS method. Using this method, the pore dimensions and the pore volume fraction were determined. The studies characterized pore sizes on length scales from 1 to 60 nm, according to the resolution of the SAXS equipment used.

The SAXS curves of the investigated cellulose membranes are shown in Figure 4 in double-logarithmic plots. It is clear that the scattering intensities of the membranes coagulated in 1-hexanol and 1-octanol were much stronger than those for other membranes. Because small-angle X-ray scattering is conditioned by the existence of the electron density inhomogeneities in the sample, which in the case of cellulose membranes is due to the existence of pores in them, the higher intensity was caused by the higher content of pores in these membranes.

The content of pores can be determined by calculating the so-called scattering power, or invariant *Q*, using the equation
(3)$$Q=\int_{0}^{\infty }q^{2}I(q)dq=\varphi (1-\varphi ){\left(\Delta \rho \right)}^{2}$$
where *I*(*q*) is the corrected SAXS intensity, *Φ* is the volume fraction of pores, and Δ*ρ* is the electron density difference between the pores and cellulose. The pores can be assumed to be entirely air-filled in dry membranes. The electron density difference between cellulose and pores was calculated to be Δ*ρ* = 511 electrons/nm^3^ [43]. The values of *Φ* calculated by Equation (3) are presented in Table 2.

The Guinier approximation can be used for the determination of pore sizes, which describes the scattering intensity for small values of the scattering vector *q*:(4)$$I(q)=I_{0}e^{-\frac{q^{2}R_{G}^{2}}{3}}$$

This equation describes the mean intensity of the radiation scattered by a particle of any shape, averaged through all its possible orientations in space. The parameter *R_G_* is the electron radius of gyration, also called Guinier’s radius. *R_G_* can be determined from the slope of the plot of ln *I(q*)-versus *q*^2^ using the intensity data in the low *q* region. An example of a Guinier plot for a membrane (B) is shown in Figure 5, and the *R_G_* values for all investigated membranes are given in Table 1.

The analysis of the results (Table 1) indicated a strong dependence of the Guinier’s radius of the pores on the molar mass of the coagulant used. Figure 6 shows that the *R_G_* increased almost linearly as the molar mass of the organic coagulant increased. Water is an inorganic compound, so it is not surprising that in this case the value of *R_G_* did not correspond to this relationship (sample W). Water is the simplest and cheapest polar solvent, and therefore it has been used as a coagulant to obtain cellulose membranes. This coagulant has the lowest molar mass and largest dipole moment of all the coagulants used in this work. The role of these two factors is discussed later in this article. The volume fraction of pores in the studied membranes also depended on the coagulant molar mass. In Figure 6, it can be seen that the pore content was almost the same in the molar mass range from 18.02 to 88.15 g/mol. The only exception was for the Me sample, for which the *Φ* considerably exceeded the average value in that range. For membranes prepared using coagulants of the highest molecular mass, the content of the pores increased rapidly, and for the membrane coagulated in 1-octanol, it was almost five times higher than in membranes obtained using coagulants of lower molar mass.

To better understand the obtained results, the mechanism of solvation of cellulose at the microscopic level should be considered. The ionic liquid used to dissolve the cellulose was a chemical compound consisting of ion-bonded anions and cations. In the liquid state, at room temperature, these solvent molecules dissociate. After mixing the cellulose and the ionic liquid, these cations and anions attack the hydrogen bonds of cellulose, tearing them apart. The disruption of hydrogen bonds results in the separation of individual cellulose chains and allows them entry into the solution. The regeneration of cellulose is based on the precipitation of the cellulose solution by the polar solvent (coagulant), which washes the ionic liquid but does not dissolve the cellulose. Washing the ionic liquid results in the formation of hydrogen bonds between cellulose chains, resulting in the recovery of the structure of the regenerated cellulose. The process of removing the solvent molecules by the coagulant molecules leads to the formation of the porous structure of the resulting cellulose. It appears that the molecular weight of the coagulant molecules had a significant impact on the size of the pores formed in this process.

The coagulation process described above is mainly based on Coulomb interactions between the molecules of the solvent and the coagulant. It therefore seems reasonable that this process will also depend on the value of the dipole moments of interacting molecules. Our research shows that the dipole moment of the coagulant was the second factor involved in determining the pore content of membranes, in addition to the molar mass. However, the effects of both these factors are complex, as illustrated by the dependence of the content of pores as a function of the dipole moment shown in Figure 7.

Undoubtedly, the content of the pores in the membrane was greatest when the values of the two factors were large, as in the case of sample Oc. If the dipole moment was large and the molar mass was small, as in the case of water, the pore content was small. The same was true in the opposite case, when the molecular weight was relatively large but the dipole moment was small, as in the case of the membrane H.

Summing up, studies have shown the influences of both the molar mass and the dipole moment of the coagulant on the content and size of the pores in cellulose membranes. However, it is difficult to definitely determine which of these factors has a crucial impact on the number of pores in the membranes. It should be emphasized that, to our knowledge, the presented studies represent the first attempts to establish the physical factors affecting these structural parameters of membranes at the molecular level, which is important from the perspective of their application.

#### 3.1.2. Morphology and Crystallinity of CEL Membranes

Cellulose is a partially crystalline polymer characterized by a polymorphism of crystallites, which can be studied by wide-angle X-ray (WAXS). WAXS investigations of CEL membranes gave an additional interesting result that completes the above-discussed SAXS findings. The native cellulose used in the membrane preparation of exhibited the cellulose I crystal structure (WAXS pattern not shown here). After dissolution and subsequent coagulation with primary alcohols, the transformation from cellulose I to the cellulose II crystalline form occurred. This phenomenon is consistent with that reported in most other known solvent systems for cellulose. The WAXS diffraction patterns (Figure 8) of the CEL membranes exhibited the characteristic cellulose II peak at 2θ = 12.6° and two peaks close to each other at 2θ = 20.3° and 21.2°, which were assigned to the (-110), (110), and (-121) crystalline planes, respectively [21].

To quantitatively examine the crystallinity of the membranes, their crystallinity indexes were evaluated from the WAXS measurements. For this purpose, each WAXS curve was deconvoluted into crystalline and amorphous scattering components using the profile fitting program WaxsFit [44]. Each peak was modeled using a Gaussian–Cauchy peak shape. The crystallinity index was calculated as the ratio of the area under the crystalline peaks to the total area of the scattering curve. The crystallinity index of native cellulose was found to be 0.756, whereas for membranes the value of this parameter ranged from 0.31 to 0.45, and this was significantly affected by the type of coagulant.

Figure 9 shows the dependence of the membrane crystallinity on the membrane pore content, and it shows that the crystallinity index decreased almost linearly as the volume fraction of the pores increased. This reflects the complexity of the process of membrane formation at the molecular level. The ionic liquid used to dissolve the cellulose broke the intermolecular and intramolecular hydrogen bonds and destroyed the primary crystalline structure. The process of removing the solvent molecules by the coagulant molecules, resulting in the recovery of the crystalline structure of cellulose, also led to the formation of its porous structure. The obtained results indicate that the formation of pores hinders the crystallization process of regenerated cellulose to some extent.

Scanning electron microscopy enabled the observation of the surface morphology of the skin layer, the support layer, and cross sections of the obtained cellulose membranes (Figure 10). The SEM images of membranes W, Et, and Pr (Figure 10-1) show cellulose layers arranged in parallel, similar to the results shown in [18]. For Me, Bu, and Pe membranes, on the other hand, a compact structure with minor inclusions was seen. The cross section of membrane He, with large, symmetrically arranged, closed chambers, was diametrically different from the other ones, whereas the SEM image of the cross section of membrane Oc was distinguished by a thin skin layer, based on the large, open, asymmetrical chambers of the support layer. In the photographs of membranes W, Me, and Et, the surface of the skin layer (Figure 10-2) was smooth, while that of membrane Pr was rough and covered with inclusions. The pictures of membranes Bu, Pe, He, and Oc reveal a very rich skin layer surface, which changed from large and flat to finer with deeper hollows. The surface of the support layers of all membranes (Figure 10-3) was richer in various morphological elements as compared to the skin layer. In the case of membrane Oc, large multilayer chambers with walls covered with micropores were observed.

#### 3.1.3. Transport Properties of CEL Membranes

An important parameter determining membrane properties is the volumetric permeate flux. Figure 10 shows that for CEL membranes, this parameter depended on the coagulant used. The lowest values of volumetric permeate flux, recorded for the membrane W for pressures of 0.1, 0.15, and 0.2 MPa, were respectively 3.33 ± 0.12; 3.87 ± 0.13, and 5.24 ± 0.39 L/m^2^ h. A slow but slight increase in the permeate flux, which was observed with the increase of working pressure, may indicate the compact structure of membrane W. It can therefore be assumed that membrane W was composed of closed pores.

The transport properties of cellulose membranes Et, Pr, and Pe were found to be similar. The permeate flux for these membranes increased slowly with the increase in working pressure, and at 0.2 MPa, their values were 9.27 ± 0.55 (membrane Et), 9.27 ± 0.37 (membrane Pr), and 7.95 ± 0.25 (membrane Pe) L/m^2^ h, respectively.

A much greater increase in the volumetric permeate flux was observed for membranes Bu and Me. Thus, as the working pressure rose, the permeate flux increased by 2–3 times for membrane Bu and 5–8 times for membrane Me. The exact values of this parameter were 4.89 ± 0.49, 10.19 ± 0.52, and 14.17 ± 0.63 L/m^2^ h (membrane Bu) and 1.25 ± 0.08, 6.22 ± 0.25, and 10.19 ± 0.46 L/m^2^ h (membrane Me).

Membranes He and Oc were characterized by very high volumetric permeate flux values. For the cellulose membrane coagulated in 1-hexanol, the following values were recorded: 15.36 ± 0.59 L/m^2^ h (for a pressure of 0.1 MPa), 42.85 ± 0.99 L/m^2^ h (for a pressure of 0.15 MPa), and 56.09 ± 1.07 L/m^2^ h (for a pressure of 0.2 MPa). The obtained results were confirmed by the high porosity of membrane He (Figure 10) observed in the SEM images, which facilitated the transport of water through the membrane. For membrane Oc, the following results were obtained: 12.72 ± 0.45 L/m^2^ h (for a pressure of 0.1 MPa), 65.70 ± 2.15 L/m^2^ h (for a pressure of 0.15 MPa), and 92.19 ± 1.51 L/m^2^ h (for a pressure of 0.2 MPa). These results show that during the operation of cellulose membrane Oc, the permeate flux increased by 5 to 7 times with the increase in working pressure. The obtained results and the SEM images (Figure 10) confirm that the cellulose membrane coagulated in 1-octanol has very good transport properties (Figure 11) because of became of large pores.

It should be noted that, as shown in Figure 11, the course of changes in the volumetric permeate flux correlated very well with the course of changes in the pore content versus the molecular mass of the coagulant, as shown above in Figure 6. The transport properties of the cellulose membranes undoubtedly depend on their porous structures, and this in turn depends on the physico-chemical properties of the coagulant used to obtain them [18,19].

#### 3.1.4. Separation Properties of CEL Membranes

It is known that cellulose-based composite membranes have been used to remove metal ions such as Fe(III) and Cu(II) [45]. A 0.1 g/dm^3^ aqueous solution of iron salt (FeCl_3_) was selected to study the separation properties of the investigated CEL membranes. A volumetric permeate flux and rejection coefficient were observed during the study (Figure 11 and Figure 12). Studies have shown that the introduction of iron ion solution into CEL membranes causes a rapid decrease in the permeate flux by approximately 67–96% at the beginning of the process. Flow through the membrane constantly decreased during membrane operation, and after about 60 min, it was about 95–98%. The obtained result may indicate that the introduction of iron ions to CEL membranes resulted in fouling on all membranes in the experiment. Kongdee et al. [46] proved that ferrous ions can be complexed on cellulose fibers, which was confirmed in our experiment.

The detailed analysis of our results showed that the highest volumetric permeate flux values during the permeation of iron ions occurred with membrane Oc. Initially, the permeate flux was 4.75 L/m^2^ h, and after 60 min, it decreased to 2.93 L/m^2^ h. For membrane Oc, the decrease in flux was 94.85–96.82% at the beginning and end of the process, respectively. At the same time, it was observed (Figure 13) that the FeCl_3_ rejection coefficient was ~61%, so membrane Oc was not clogged completely. This result confirms that it was very porous with large pores. Similar results were obtained for the study of transport properties of membrane He for which the volumetric permeate flux values were in the range of 2.28–1.14 L/m^2^ h (Figure 12). The decrease in the permeate flux was 95.94–97.97% relative to that obtained for distilled water. The FeCl_3_ rejection coefficient on membrane He was 62.32%, confirming the high porosity of this membrane. A similar rejection coefficient result of 61.31% was obtained for membrane W (Figure 13). At the same time, it was observed that the decrease of the volumetric rejection coefficient at the beginning of the process was the lowest (66.78%) during the flow of iron(III) ions through membrane W, and this increased to 94.46% after 60 min. The obtained results suggest that membranes Oc, He, and W, characterized by low density and high porosity, had similar separation properties relative to the aqueous solution of FeCl_3_.

For the remaining cellulose membranes (Me, Et, Pr, Bu, and Pe), the decrease in the specific permeate flux for iron ions was similar and ranged between 95% and 98%, with a rejection coefficient of ~81.5% (for membrane Bu), ~84% (for membrane Pe), ~85.5% (for membrane Et), ~86% (for membrane Me), and 86.9% (for membrane Pr). These membranes almost completely blocked iron(III) ions, while the permeate flux decreased to very low values of 0.15–0.34 L/m^2^ h. Membranes Me, Et, Pr, Bu, and Pe could be used to remove metals from solutions, as suggested in the work of Cifci et al. [45]. The phenomenon of removal of Fe(III) ions on cellulose membranes may be a result of the formation of chemical bonds between metal ions and the functional groups of cellulose. Yadav [33] used the phenomenon of Fe ion complexation on cellulose, obtaining new, previously unmentioned bionanocomposites.

### 3.2. Graphene Oxide/Cellulose (GO/CEL) Membranes

Graphene oxide (GO) is a modern material used for the production of polymer composites. GO has many different oxygen-containing functional groups, such as epoxy, hydroxyl, carbonyl, and carboxyl [47,48]. Oxygen groups give hydrophilic properties to graphene oxide, making it easy to form stable aqueous dispersions [49,50] at concentrations above 3 mg/mL [51]. GO is also used in membranes, along with carbon nanotubes [52,53,54,55] and graphene [56,57,58]. It allows thin monolayer films to be obtained [54,55,56,57], which can be used for desalination and purification [56,57,59,60] as well as membrane distillation [61].

The GO/CEL membranes are composites in which cellulose chains containing hydroxyl groups form hydrogen bonds with oxygen functional groups on the surfaces and edges of GO flakes (Figure 14) [38,62]. 

For the preparation of GO/CEL membranes, a method of combining the cellulose solution with a GO dispersion was developed to obtain a homogeneous solution from which the composite membranes were then formed using the phase inversion method by coagulation in distilled water. As a result, cellulose composite membranes (labeled A–F) which differed in color depending on the amount of GO added were obtained (Figure 15). A pure cellulose membrane (labelled “0”) was also obtained to compare its properties with the properties of membranes containing GO. As shown in Figure 14, membrane “0” was colorless, while the other diaphragms were shades of grey. The -bigger amount of GO was added the darker the GO/CEL composite membranes became.

#### 3.2.1. Morphology and Structure of GO/CEL Membranes

In the SEM images of membrane cross sections (Figure 1601–F1), the pure cellulose membrane (“0”) was observed to be compact but slightly rough, which made it clearly distinguishable from the GO/CEL composite membranes. The cross sections of membranes A and B were similar to each other. In the case of composite membranes A and B, apart from the roughness, horizontal cracks appeared. Further, the addition of 0.1% and 0.2% GO increased the thickness of the membranes (Figure 16A1,B1). The cross-sectional structures of membranes C and D were different from all others. Membranes containing 1% and 2% GO had compact structures but with visible horizontal cracks (Figure 16C1,D1). By comparing the thickness of the membranes, it was seen that a 1% addition of GO to composite membrane C reduced its thickness, which was comparable with the thickness of the “0” membrane. Many horizontal cracks were observed in the SEM images of membrane E. The structure of can be observed as being entirely composed of transversely arranged flakes. The effect of GO addition on the cellulose matrix was easily observed using the microscopic observations of membrane cross sections. The effect of increasing the amount of GO in the composite membrane was the appearance of an increasing number of cracks, which may have resulted from the arrangement of components in the composite. Phiri et al. also observed the formation of layers in the cross sections of composite membranes based on microfibrillated cellulose with the addition of GO [63].

The morphology of the skin layers of the membranes can be observed in Figure 1602–F2. Membrane “0” was characterized by an even, smooth surface, while the surfaces of composite membranes with small amounts of added GO (membranes A, B, C) were very rich in corrugations, spherical beads, and recesses. In contrast, the skin layers of membranes D, E, and F were different from the others. On the relatively flat surface of these membranes, cracks and recesses were observed, the number and size of which increased with the amount of GO added in the direction of membranes D to F.

The surface of the support layers of all membranes (Figure 1603–F3) was richer in various morphological structure elements compared with the skin layer. Hollows and spherical beads were observed, the number and size of which increased with the amount of added GO. In the case of membranes E and F, apart from hollows, numerous cracks appeared on the surface of the support layer.

Figure 17 shows the WAXS diffraction curves of membrane “0” (the pure cellulose) and the pure GO. As described in Section 3.1.2 (Figure 8), the cellulose curve contains all peaks characteristic for cellulose II. The WAXS curve of the GO shows a sharp diffraction peak at 2θ = 9.0° (d-spacing = 0.98 nm), which reflects the gallery gap of the layered structure of the GO.

Figure 18 presents the WAXS diffraction patterns of the GO/CEL membranes. The angular position of the GO peak was shifted towards the lower diffraction angles with a decreasing GO content. This peak shift indicates an increase in the distance between the carbon layers in the GO. Compared to pure GO, this distance increased from 0.98 to 1.6 nm for membranes B and D, respectively. The stepwise increase in d-spacing might be due to the interactions between GO and cellulose by hydrogen bonds (Figure 14), which might have already been saturated at a low loading of the GO. Therefore, the increase in the distance between carbon galleries in GO was more pronounced when the ratio of cellulose content to GO content increased.

#### 3.2.2. Transport Properties of GO/CEL Membranes

In order to investigate the effect of adding GO on the transport properties of cellulose membranes, we determined the permeate water flux for the obtained membranes. The course of changes in the value of this parameter for individual membranes is illustrated in Figure 19. The lowest values of volumetric permeate flux were recorded for membrane “0”, and for the subsequent pressures of 0.1, 0.15, and 0.2 MPa, they were respectively 2.93 ± 0.02, 3.76 ± 0.07, and 4.36 ± 0.03 L/m^2^ h. A slow but slight increase in the permeate flux which was observed with the increase of working pressure suggests that membrane “0” had a compact structure. It can therefore be assumed that membrane “0” was composed of closed pores, as confirmed by SEM images (Figure 1601).

The transport properties of the composite membranes A, B, and C were similar. The permeate flux for these membranes increased slowly with the increase in working pressure, and at 0.2 MPa, it was 4.58 ± 0.32 (membrane A), 5.26 ± 0.10 (membrane B), and 5.43 ± 0.07 (membrane C) L/m^2^ h. The results show that the addition of 0.1%, 0.2%, and 1.0% *w/w* of GO to the cellulose only slightly improved the transport properties of the obtained GO/CEL membranes. In the case of membranes A and B, it can be assumed that the structures observed in SEM images contained mainly closed pores. However, in the case of membrane C, low values of the permeate water flux may have been a result of the formation of hydrogen bonds between GO molecules and cellulose chains (Figure 14), which results in hydrophobic membranes, as described in our earlier publication [64]. This could be confirmed by the low thickness of membrane C observed in SEM images (Figure 16C1).

An increase in the distilled water flow through cellulose composite membranes was observed for GO added at concentrations of 2%, 10%, and 20% *w/w*. The permeate volumetric flux values for the working pressure of 0.2 MPa were 10.24 ± 0.22 (membrane D), 21.71 ± 0.68 (membrane E), and 40.20 ± 2.33 (membrane F) L/m^2^ h, respectively. The results show that the addition of graphene oxide to cellulose in amounts of 2%, 10%, and 20% *w/w* improved the transport properties of the obtained GO/CEL membranes by 2, 5, and 10 times in comparison to membrane “0”. Thus, it can be assumed that due to the large amount of GO, membranes D, E, and F had layered structures (Figure 16D1,E1,F1) facilitating the transport of water despite the high thickness of these composite membranes.

#### 3.2.3. Separation Properties of GO/CEL Membranes

The present study also examined whether the obtained membranes could be used to remove bivalent heavy metal ions from aqueous solutions. All obtained membranes—both pure cellulose and cellulose with added components—were characterized by high rejection coefficient (R) values. Membrane “0” (Figure 20a) was characterized by rejection coefficients of 100% (Co), 93% (Zn), ~69% (Pb), 40% (Ni), and 1% (Na). On the other hand, the studies carried out on the synthetic wastewater mixture (Figure 20b) showed that the R coefficients for the pure cellulose membrane were 100% (Co), ~ 87% (Zn), ~71% (Pb), ~79% (Ni), and ~2% (Na). It was observed that in the mixture of synthetic wastewater, preferential rejection of nickel ions on membrane “0” and an increase in the degree of lead ion rejection with respect to zinc ions, for which the R coefficient decreased, occurred.

All cellulose GO/CEL composite membranes studied in the experiment were characterized by higher rejection coefficient values than membrane “0” (Figure 20). Thus, the nanoaddition of GO in the cellulose matrix increased the amount of heavy metal ions deposited on the membranes. By analyzing the results obtained for membrane D, it was observed that the addition of 2% *w/w* of graphene oxide resulted in the highest R coefficient values of 100% (Co); 98–100% (Zn); 97–99% (Ni); and 69–71% (Pb). An increase in the amount of GO addition to 10% or 20% did not improve the degree of ion rejection on membranes; rather, it slightly decreased it by 1%.

The explanation of the observed phenomenon may be that the concentration of GO in the composite was too high, resulting in the formation of layers and cracks, which were observed in cross sections (Figure 16D1,E1,F1). We do not know exactly how the cellulose and GO layers were arranged in our composite. However, Chen et al. [65] showed that the arrangement of the electronegative and electropositive components of the composite is responsible for the separation properties of the membranes, as well as the order in which heavy metals are removed from solution.

The results of this study show that the membranes obtained in the experiment were characterized by very high rejection coefficients in relation to the divalent metal ions cobalt, zinc, nickel, and lead. For Co (II) ions, total rejection was observed on each studied membrane, both for the solution containing single ions and for the synthetic wastewater mixture. In the case of other ions, on the other hand, the degree of rejection (R) of divalent ions was in the range of 85–99%. The low rejection coefficient for sodium ions (1–2%) indicates that we obtained an ultrafiltration membrane.

The observed changes may be the result of interactions between functional groups of cellulose chains and metal ions. The cellulose membrane is negatively charged, and the charge is derived from hydroxyl groups (primary and secondary), which, according to Lewis theory, are hard bases. Co^2+^, Ni^2+^, Zn^2+^, and Pb^2+^ ions, on the other hand, are indirect acids. Thus, the formation of chemical (ionic and hydrogen) bonds between metal ions and cellulose chains is possible. The resulting combination rejects heavy metal ions on the membrane but does not cause fouling. The increase in the permeate flow, on the other hand, may result from the formation of channels through which small particles of water flow freely.

## 4. Conclusions

This paper presents the results of research on the production of cellulose membranes from cellulose solution in EMIMAc ionic liquid. The membranes were formed by phase inversion using water and primary alcohols as coagulants. The results of structural investigations performed by X-ray methods (WAXS and SAXS) reflected the complexity of the membrane formation process at the molecular level. Our research shows that the physicochemical properties of the coagulant used, such as molecular mass and dipole moment, play important roles in this process. It was found that both the content and dimensions of the pores depended on the molecular mass of the coagulant used. It was also found that the dipole moment of coagulant molecules had a large influence on the volume content of the pores. The highest content of pores was obtained when these two factors represented a substantial value. In cases where any of these factors was of little value, the pore content decreased significantly. The discussed studies also revealed the effect of the pore content on the crystallinity of cellulose membranes; namely, the degree of crystallinity decreased almost linearly as the volume fraction of pores increased. This indicates that the formation of pores hinders the crystallization process of regenerated cellulose to some extent.

The porous structure of CEL membranes has a significant impact on their transport properties. The lowest values of the volumetric permeate flux occurred for membrane (W), which was formed using the coagulant of the lowest molecular mass, while the highest value of this parameter occurred for the membranes (He and Oc) coagulated using coagulants with large molecular masses.

Studies of the separation properties of CEL membranes carried out using iron(III) solution showed a very high decrease in flow of 67–96% and a high rejection coefficient (82–87%), indicating a lack of resistance to the unfavorable phenomenon of fouling.

The introduction of GO into the cellulose matrix influenced the process of membrane formation and, consequently, the physicochemical, transport, and separation properties of the GO/CEL composite membranes. Structural studies have shown that the incorporation of a small GO additive into the cellulose membranes results in a homogeneous composite in which the individual components are connected by hydrogen bonds. The use of SEM and WAXS showed that in the case of a relatively high GO content, the excess admixture separated and an isolated GO phase occurred in the GO/CEL membrane structure, and this was responsible for very good transport and separation membrane properties. The addition of graphene oxide improved the flow of water through membranes by up to 10 times. For these reasons, it is advisable to add GO to the CEL matrix. Composite membranes containing 2%, 10%, and 20% GO also had very good rejection of heavy metal ions. Due to the bactericidal GO admixture, GO/CEL membranes could also find a potential application in water disinfection [66].

## Figures and Tables

**Figure 1 polymers-11-01178-f001:**
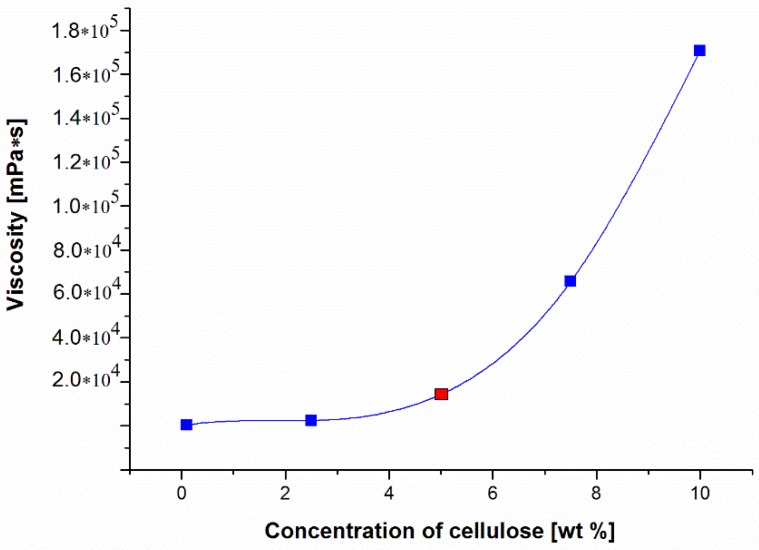
The results of viscosity tests of cellulose solutions in 1-ethyl-3-methylimidazolium acetate (EMIMAc) (measurement temperature 25 °C).

**Figure 2 polymers-11-01178-f002:**
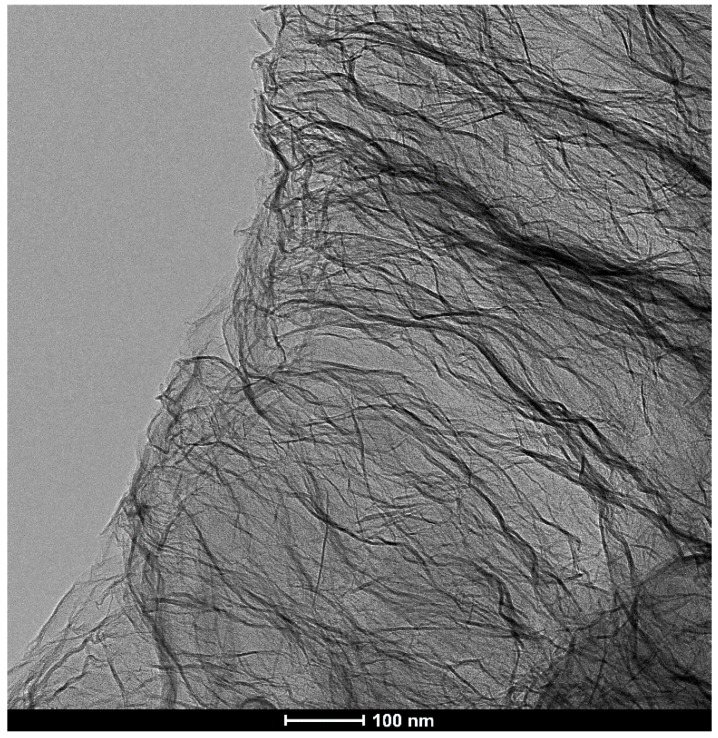
TEM (transmission electron microscopy ) image of graphene oxide (GO) flakes.

**Figure 3 polymers-11-01178-f003:**
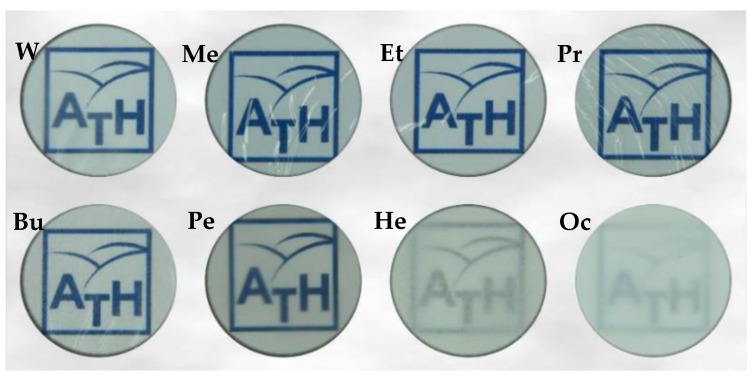
Images of a pure cellulose membrane coagulated in distilled water (W), methanol (Me), ethanol (Et), 1-propanol (Pr), 1-butanol (Bu), 1-pentanol (Pe), 1-hexanol (He) and 1-octanol (Oc) (our university’s logo was placed under the membranes for a better display of their transparency).

**Figure 4 polymers-11-01178-f004:**
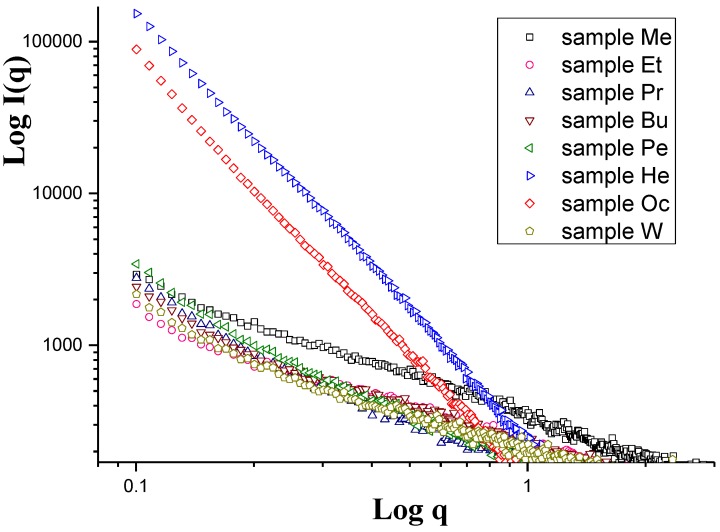
Double-logarithmic small-angle X-ray scattering (SAXS) curves of cellulose membranes.

**Figure 5 polymers-11-01178-f005:**
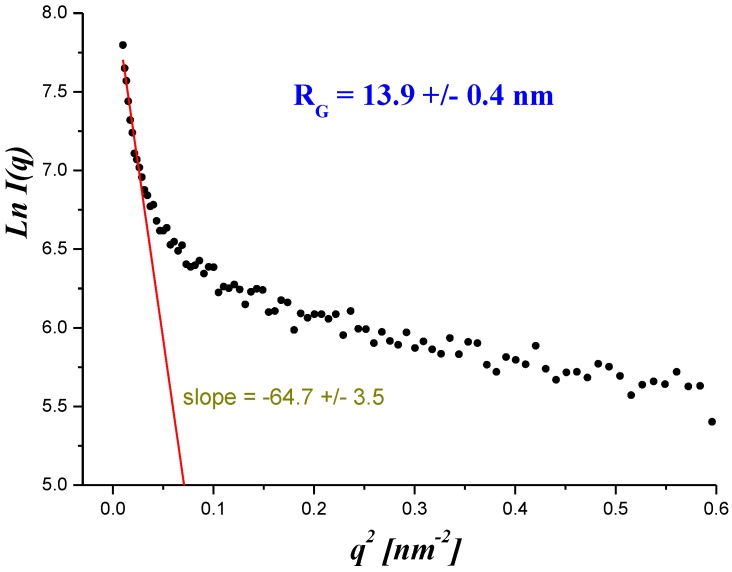
The Guinier plot for the Bu sample.

**Figure 6 polymers-11-01178-f006:**
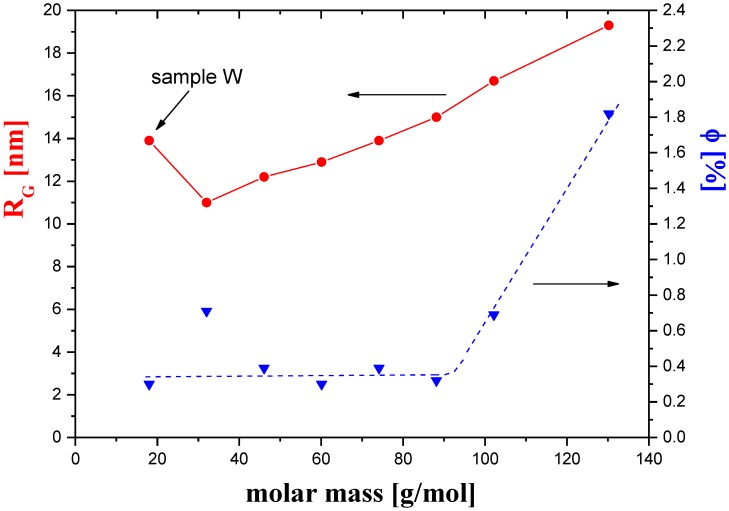
The radius of gyration (*R_G_*) and the volume fraction of pores (*Φ*) as a function of the coagulant molar mass.

**Figure 7 polymers-11-01178-f007:**
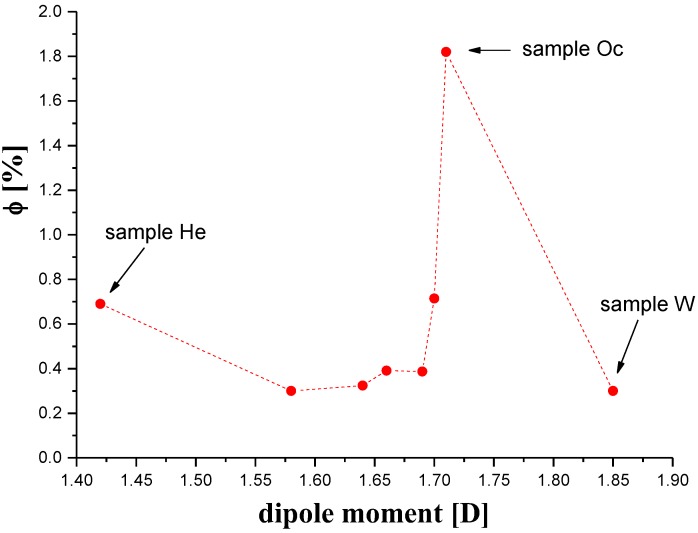
The volume fraction of pores (*Φ*) as a function of the coagulant dipole moment.

**Figure 8 polymers-11-01178-f008:**
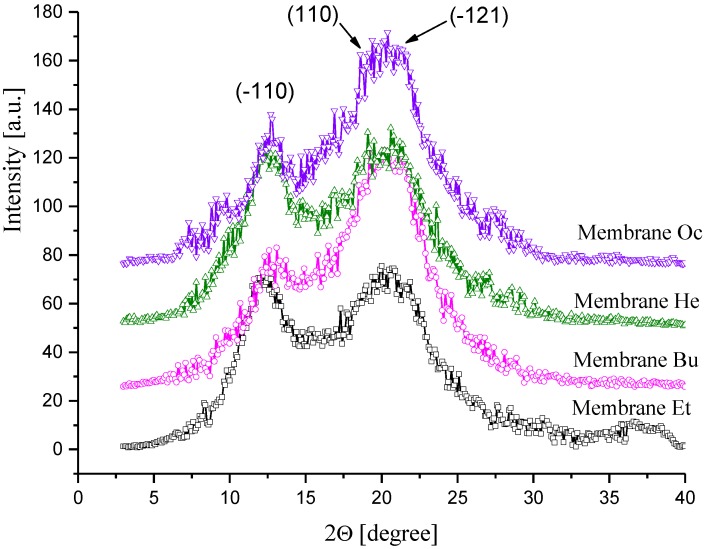
Examples of wide-angle X-ray (WAXS) patterns of CEL membranes.

**Figure 9 polymers-11-01178-f009:**
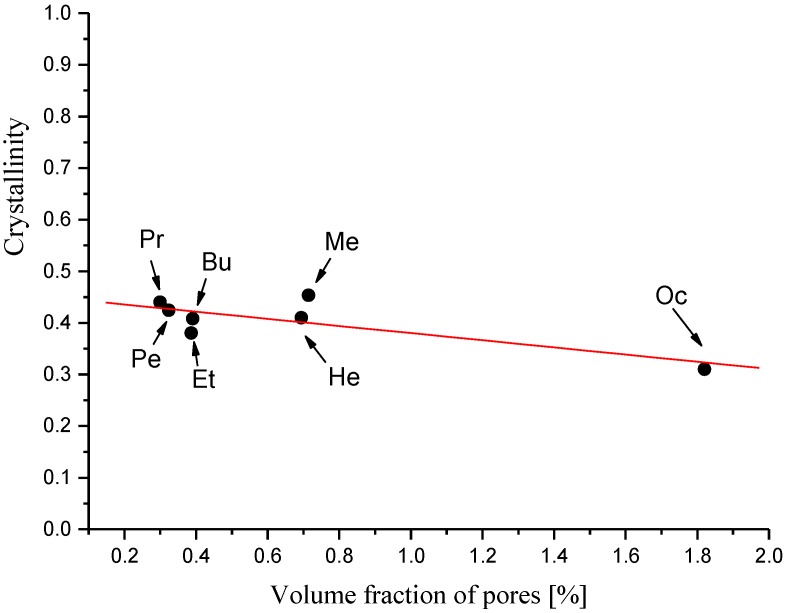
Effect of the pore volume fraction on the membrane crystallinity.

**Figure 10 polymers-11-01178-f010:**
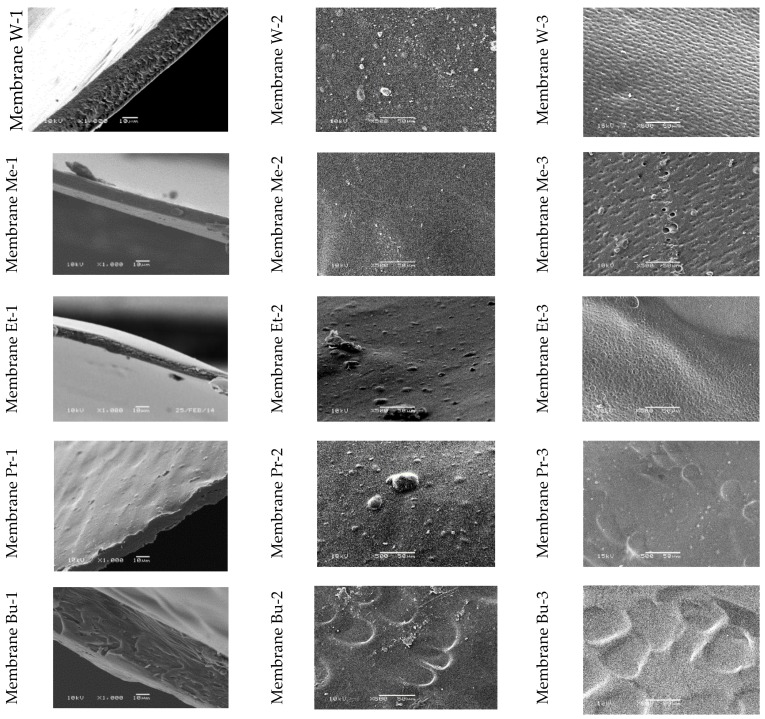
SEM images for cellulose membranes: (**1**) cross-section; (**2**) skin layer; (**3**) bottom layer.

**Figure 11 polymers-11-01178-f011:**
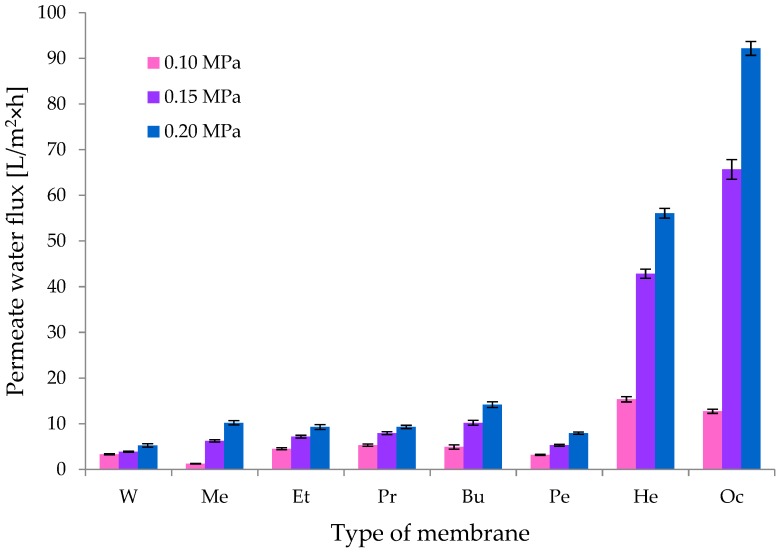
Water flux for the obtained cellulose membranes.

**Figure 12 polymers-11-01178-f012:**
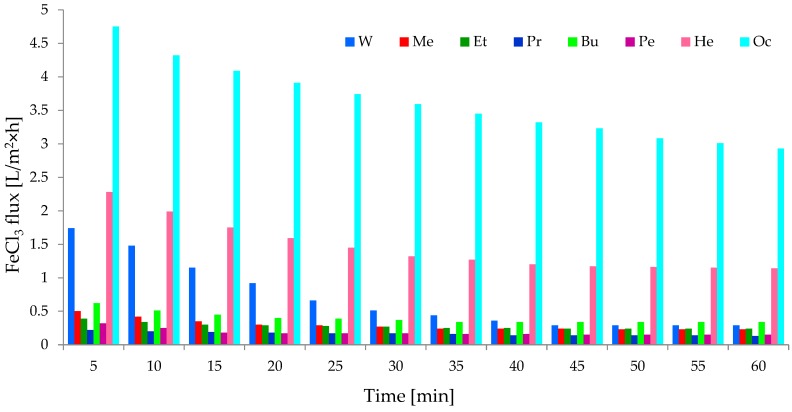
Volumetric permeate flux for the solution of FeCl_3_.

**Figure 13 polymers-11-01178-f013:**
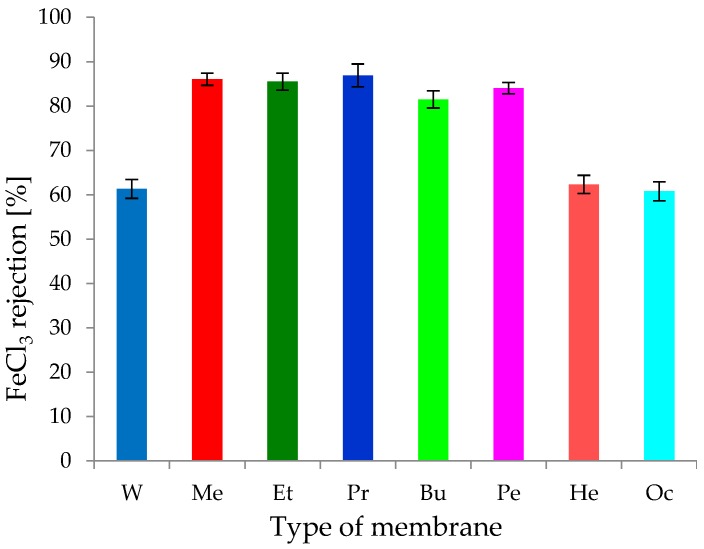
Rejection coefficient (R) for FeCl_3_ solution.

**Figure 14 polymers-11-01178-f014:**
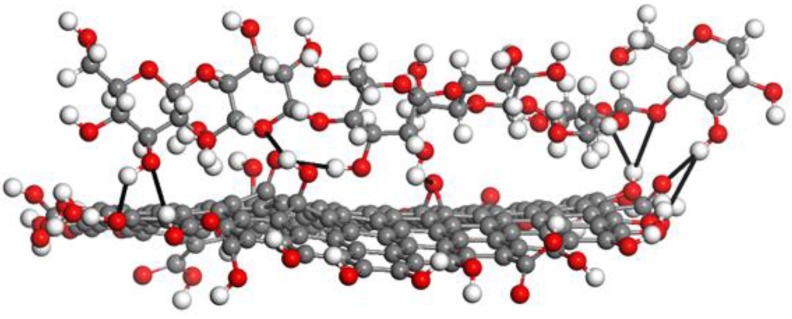
Schematic diagram of hydrogen bonds (black bold line) between the GO (bottom) and CEL macromolecular chains (top). Grey balls—carbon, white balls—hydrogen, red balls—oxygen.

**Figure 15 polymers-11-01178-f015:**
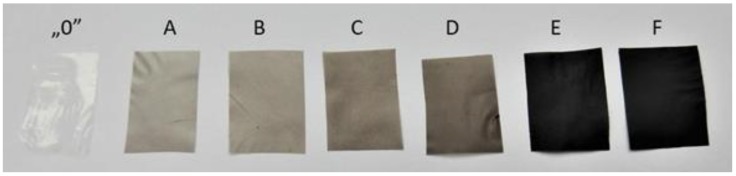
Images of a pure cellulose membrane (“0”) and GO/CEL composite membranes (A—0.1% GO, B—0.2% GO, C—1% GO, D—2% GO, E—10% GO, F—20% GO).

**Figure 16 polymers-11-01178-f016:**
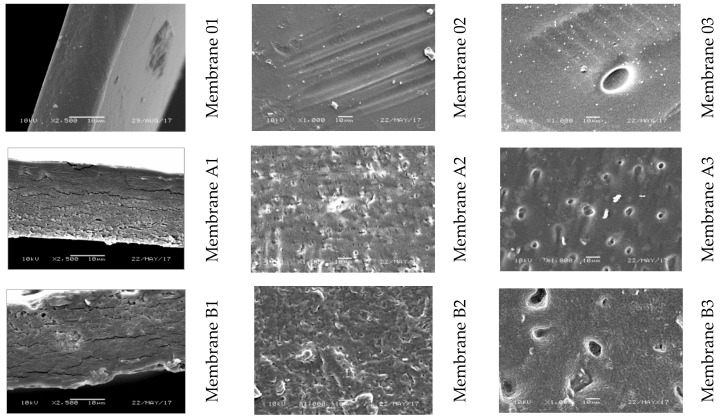
SEM images for the pure cellulose membrane and GO/CEL composite membranes: (**1**) cross section; (**2**) skin layer; (**3**) bottom layer.

**Figure 17 polymers-11-01178-f017:**
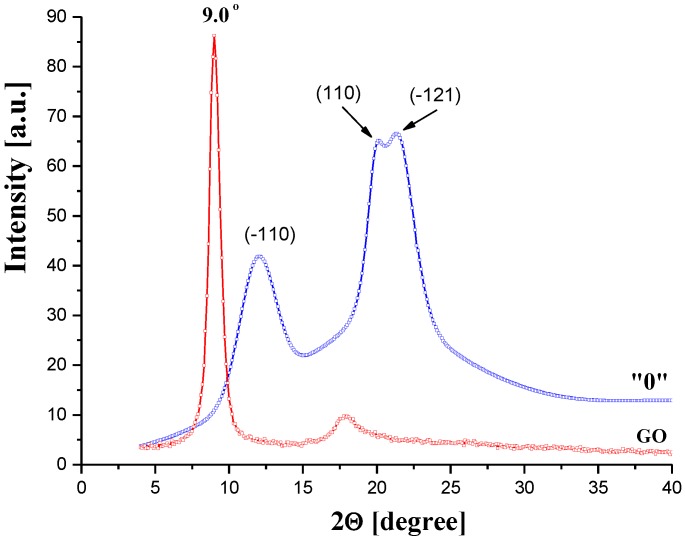
WAXS patterns of membrane “0” (pure cellulose) and pure GO.

**Figure 18 polymers-11-01178-f018:**
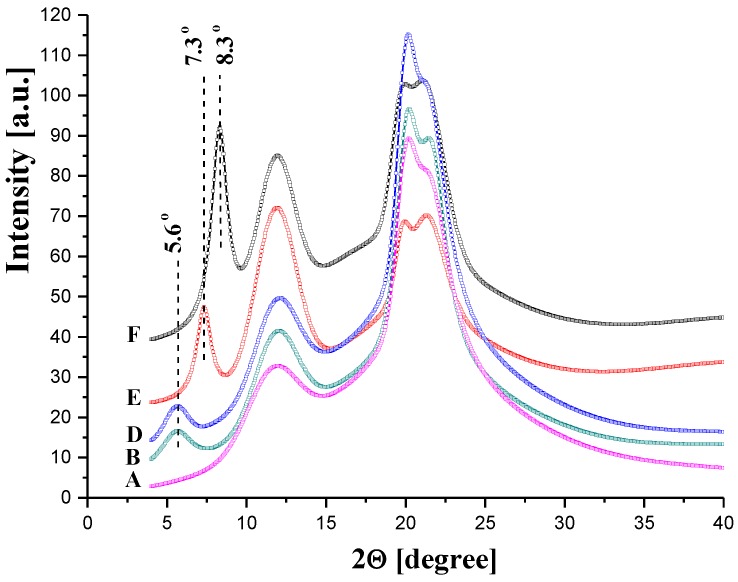
Examples of WAXS patterns of cellulose membranes (the letters on the left indicate the membrane codes).

**Figure 19 polymers-11-01178-f019:**
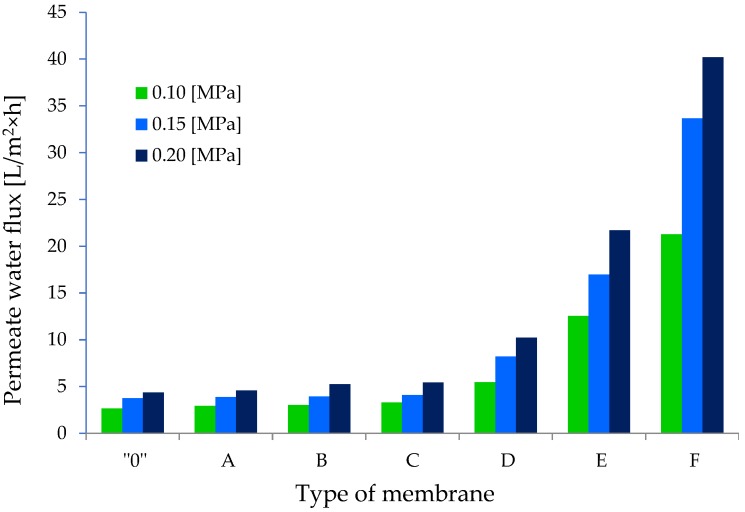
Pure water permeability for the pure cellulose membrane and GO/CEL composite membranes.

**Figure 20 polymers-11-01178-f020:**
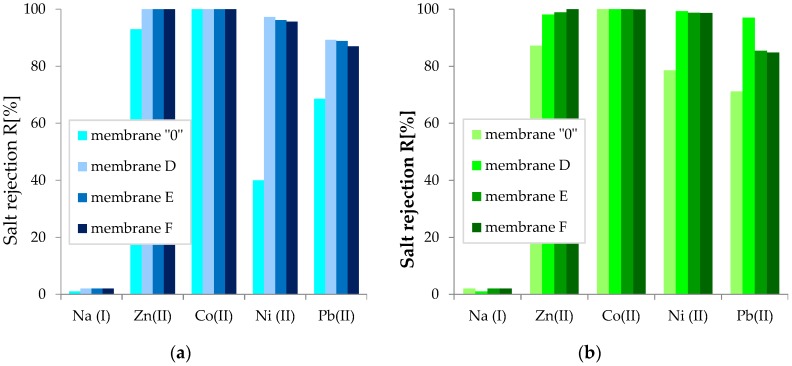
Rejection coefficients on pure cellulose membrane and GO/CEL composite membranes of (**a**) single metals; (**b**) subsequent metals in synthetic wastewater.

**Table 1 polymers-11-01178-t001:** The composition of the solutions for the preparation of membranes. CEL: pure cellulose; DMF: *N*,*N*-dimethylformamide.

Membrane Designation	“0”	A	B	C	D	E	F
**Amount of 5.2% GO/DMF Solution (g)**	0	0.05	0.1	0.5	0.97	4.8	9.7
**Amount of CEL (g)**	2.5	2.5	2.5	2.5	2.5	2.5	2.5
**Amount of EMIMAc (g)**	47.5	47.45	47.4	47.0	46.53	42.7	37.8
**Concentration of GO (% *w/w*)**	0	0.1	0.2	1	2	10	20
**Concentration of CEL (% *w/w*)**	100	99.9	99.8	99	98	90	80

**Table 2 polymers-11-01178-t002:** Characteristics of the pure cellulose (CEL) membranes used for the study and values of structural parameters obtained by means of the SAXS method. *R_G_*: radius of gyration, *Φ*: volume fraction of pores.

Sample	Coagulant			SAXS Results	
	Type	Molar Mass (g/mol)	Dipole Moment (D)	*R_G_* (nm)	*Φ* (%)
Me	methanol	32.04	1.70	11.0 ± 0.3	0.71
Et	ethanol	46.07	1.69	12.2 ± 0.2	0.39
Pr	1-propanol	60.1	1.58	12.9 ± 0.3	0.30
Bu	1-butanol	74.12	1.66	13.9 ± 0.4	0.39
Pe	1-pentanol	88.15	1.64	15.0 ± 0.4	0.32
He	1-hexanol	102.17	1.42	16.7 ± 0.3	0.69
Oc	1-octanol	130.23	1.71	19.3 ± 0.5	1.82
W	water	18.02	1.85	13.9 ± 0.3	0.30

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
