# Peer review of "Structure–Property Relationships of Pure Cellulose and GO/CEL Membranes Regenerated from Ionic Liquid Solutions"

_polymers, 2019, doi:10.3390/polym11071178_

Reviewer 1 Report

This study characterized the cellulose and cellulose-graphene oxide membranes in terms of their pore structure, crystallinity, morphology, and ability to absorb heavy metals. The membranes were prepared in ionic solvent and coagulated in water and different alcohol solvents. This is an interesting work, but there are some items that need to be addressed properly.

-        Although the cellulose membrane prepared through a green method, an environmentally unfriendly solvent (dimethylformamide, DMF) applied for dispersion of graphene oxide. It is very clear that the dispersion of GO in other solvents such as water required stabilizer in order to avoid any agglomeration, but the environmental impact of such solvent would have been much lower than the DMF.   

-        Preparations of cellulose membrane and GO/CEL membrane were exactly the same. It is not needed to repeat the same sentences two times.

-        There is no information about the size and structure of GO.

-        In part 2.6, the authors explain the measurement of water flux and mentioned that the dry membranes were immersed in distilled water. How the membranes were dried (freeze-dried or oven dried, what was the temperature)? What was the temperature of distilled water?

-        How the SEM analyses were performed

-        The sample codes are misleading, for instance, there is a sample with label A-F for GO/CEL membrane and also another sample in ethanol and 1-butanol for CEL membrane, which were named as B and E.

-        There is no literature support in some of the results and discussion part such as in “Porous structure of CEL membranes”, “Morphology and crystallinity of CEL membranes”, “Transport properties of CEL membranes”, “Morphology and structure of GO/CEL membranes”

-        In general, the results and discussion part must be improved by referring to more relevant literature.

-        In conclusion, it must be mentioned whether the authors suggest the use of GO in the membrane or not?

-        Reference number 32 was not cited in the text.

Reviewer 2 Report

Reviewer’s comments:

Ślusarczyk group prepared two types of cellulose membranes by classical wet phase inversion method from a solution of the polymer in 1-ethyl-3-methylimidazolium acetate, by coagulation in water and selected primary alcohols. Studies are well but not enough for publication. It needs a lot of changes.

1.      Author used GO in experiment with cellulose but title of the paper is not verifying. Please try to insert GO and cellulose related something in it.

2.      Abstract part should be quantitative and try to insert the results. You can see some published papers.

3.      Introduction part is not attractive and try to update by recent papers (2016, 2017,2018 & 2019).

4.      Insert reference for equation (1) and (2)

5.      Line from……………….“The transport properties of cellulose membranes undoubtedly depend on their porous structure, and this in turn depends on the physico-chemical properties of the coagulant used to obtain them”. Needs current references.

6.      Section 3.1.4 needs some current references.

7.      Section 3.2 should insert an important paper based on GO/CMC. Check Cellulose 20 (2), 687-698 and cite it.

8.      ……………….Give reference for “The WAXS curve of the GO shows the sharp diffraction peak at 2θ = 9.0° (d-spacing = 0.98 nm) which reflects the gallery
gap of the layered structure of the GO”. Check it and modify.

9.      Conclusion should be concise. I think it is too big. Check and modify it.

10.  In result and discussion part, some refrences should belong by year 2018 and 2019.

11.  Possible to insert the some refrences regarding GO and cellulose. Carbohydrate polymers 110, 18-25, Composites Communications 10 (1), 1-5

12.  Reported characterization should be insert in details. The mistakes in English should be rechecked and modify it.

13.  Introduction, result and discussion part should be updated by 2019 research paper.

14.  Possible, insert the TEM of fabricated of GO.

Author Response

Round  2

Reviewer 1 Report

I have no further comment.  

Reviewer 2 Report

Accepted.